

# Gamma-aminobutyric acid elicits $H_2O_2$ signalling and promotes wheat seed germination under combined salt and heat stress

Song Yu[1], Zhihan Lian[1], Lihe Yu[1,2], Wei Guo[1,2], Chunyu Zhang[1,2] and Yifei Zhang[1,2]

[1] Department of Agronomy and Crop Sciences, College of Agriculture, Heilongjiang Bayi Agricultural University/Heilongjiang Provincial Key Laboratory of Modern Agricultural Cultivation and Crop Germplasm Improvement, Daqing, Heilongjiang Province, China
[2] Key Laboratory of Low-carbon Green Agriculture in Northeastern China, Ministry of Agriculture and Rural Affairs, Daqing, Heilongjiang Province, China

Corresponding authors
Lihe Yu, yulihe2002@126.com
Yifei Zhang, byndzyf@163.com

## ABSTRACT

**Background:** In the realm of wheat seed germination, abiotic stresses such as salinity and high temperature have been shown to hinder the process. These stresses can lead to the production of reactive oxygen species, which, within a certain concentration range, may actually facilitate seed germination. γ-aminobutyric acid (GABA), a non-protein amino acid, serves as a crucial signaling molecule in the promotion of seed germination. Nevertheless, the potential of GABA to regulate seed germination under the simultaneous stress of heat and salinity remains unexplored in current literature.

**Methods:** This study employed observational methods to assess seed germination rate (GR), physiological methods to measure $H_2O_2$ content, and the activities of glutamate decarboxylase (GAD), NADPH oxidase (NOX), superoxide dismutase (SOD), and catalase (CAT). The levels of ABA and GABA were quantified using high-performance liquid chromatography technology. Furthermore, quantitative real-time PCR technology was utilized to analyze the expression levels of two genes encoding antioxidant enzymes, MnSOD and CAT.

**Results:** The findings indicated that combined stress (30 °C + 50 mM NaCl) decreased the GR of wheat seeds to about 21%, while treatment with 2 mM GABA increased the GR to about 48%. However, the stimulatory effect of GABA was mitigated by the presence of ABA, dimethylthiourea, and NOX inhibitor, but was strengthened by $H_2O_2$, antioxidant enzyme inhibitor, fluridone, and gibberellin. In comparison to the control group (20 °C + 0 mM NaCl), this combined stress led to elevated levels of ABA, reduced GAD and NOX activity, and a decrease in $H_2O_2$ and GABA content. Further investigation revealed that this combined stress significantly suppressed the activity of superoxide dismutase (SOD) and catalase (CAT), as well as downregulated the gene expression levels of *MnSOD* and *CAT*. However, the study demonstrates that exogenous GABA effectively reversed the inhibitory effects of combined stress on wheat seed germination. These findings suggest that GABA-induced NOX-mediated $H_2O_2$ signalling plays a crucial role in mitigating the adverse impact of combined stress on wheat seed germination. This research holds

significant theoretical and practical implications for the regulation of crop seed germination by GABA under conditions of combined stress.

# INTRODUCTION

Plants, being sessile organisms, are vulnerable to a range of abiotic stresses in their natural environment, including high temperatures and salinity, which can impact their growth across their entire life cycle (*Kumar, 2020*; *Rankenberg et al., 2021*). These abiotic stresses can trigger an excessive prodiction of reactive oxygen species (ROS), leading to significant hindrances in normal plants growth (*Sachdev et al., 2021*; *Wang et al., 2024*). To counteract oxidative stress, plants rely on enzymatic antioxidants such as superoxide dismutase (SOD, EC 1.15.1.1) and glutathione to mitigate the effects of ROS (*Sachdev et al., 2021*; *Wang et al., 2024*). However, it is important to note that low concentrations of ROS, specifically hydrogen peroxide ($H_2O_2$), can also have a beneficial impact. For instance, the ROS generated by NADPH oxidase (NOX, EC 1.6.99.6) serves as a vital signalling molecule in the plant's response to stress (*Hu et al., 2020*). An imbalance in NOX activity, whether excessive or deficient, can hinder seed germination (*Ishibashi et al., 2010*; *Zhang, Deng & Li, 2018*).

The involvement of ROS, particularly $H_2O_2$, is essential in the germination process, likely due to the elevated levels of abscisic acid (ABA) induced by dormancy, which necessitates ROS for degradation (*Liu et al., 2010*; *Cheng et al., 2022*). This phenomenon underscores the detrimental impact of low ROS concentration on seed germination. Additionally, seed germination is influenced by adverse stressors such as elevated temperatures and salt damage (*Lei, Song & Fu, 2005*; *Nunes et al., 2019*). While abiotic stressors may induce excessive ROS production in seeds (*Zhang, Shi & Deng, 2018*), they can also trigger the activation of ABA signalling pathways, ultimately halting the germination process (*Liu et al., 2019*). Consequently, ROS exhibit dual functions in seed germination, necessitating maintenance within an optimal concentration range for successful germination (*Bailly, El-Maarouf-Bouteau & Corbineau, 2008*; *Bailly, 2019*).

When plants face multiple environmental stresses, studies have shown that the effects of different environmental stresses on plants are not simply superimposed (*Flores, Pérez-Sánchez & Jurado, 2017*; *Zandalinas et al., 2021*; *Zandalinas & Mittler, 2022*; *Zandalinas et al., 2024*). For example, high temperatures can further exacerbate the negative effects of salt damage on *Jatropha curcas*, affecting stomatal opening, $CO_2$ absorption, $Na^+$ accumulation, antioxidant enzyme activity, and $H_2O_2$ content in the antioxidant defence system (*Silva et al., 2013*).

Abiotic stresses such as drought, high temperature, and salt damage significantly impact wheat, exceeding the harm caused by biotic stresses (*Abhinandan et al., 2018*; *Dhakal et al., 2021*). These stresses affect various stages from seed germination and seedling growth to

grain yield (*Miransari & Smith, 2019*; *Khaeim et al., 2022*). For seed germination, wheat seeds require appropriate water, temperature, and ventilation (*Lafond & Fowler, 1989*; *Jame & Cutforth, 2004*).

However, seed germination often faces multiple environmental stressors simultaneously (*Lei, Song & Fu, 2005*; *Khaeim et al., 2022*; *Sartori et al., 2023*).

Salinity can induce the accumulation of free amino acids in germinated seeds (*Dubey & Rani, 1989*). Numerous studies have demonstrated that both protein and non-protein free amino acids play a crucial role in seed germination (*Kuo et al., 2004*; *Alhadi et al., 2012*; *Amir, Galili & Cohen, 2018*; *Begum et al., 2022*). For example, the germination of soybean seeds increases the content of the non-protein amino acid γ-aminobutyric acid (GABA) (*Kuo et al., 2004*), which is produced from the decarboxylation of L-glutamic acid catalysed by glutamate decarboxylase (GAD, EC 4.1.1.15). Moreover, GABA activates α-amylase activities in seeds and promotes seed germination under both favourable and saline conditions (*Cheng et al., 2018*; *Sheng et al., 2018*). However, there have been no reports yet on whether GABA can also enhance seed germination under combined stresses. In the current research, we aim to use GABA to enhance wheat seed germination under the combined stress of salt and high temperature, and to elucidate the underlying mechanisms from the perspectives of ROS and ABA accumulation. This work will provide new insights for improving crop seed germination in the field.

## MATERIALS AND METHODS

### Reagent preparation

The reagents GABA, ABA, fluridone (Flu), gibberellin (GA), dimethylthiourea (DMTU), imidazole (IMZ), diethyldithiocarbamic acid (DDC), aminotriazole (ATZ), and diphenyleneiodonium chloride (DPI) were purchased from Macklin Biochemistry & Technique Company (Shanghai, China). The reagents and their respective concentrations, including GABA (0.5, 2, and 10.0 mM), ATZ (2 mM), Flu (0.1 mM), GA (0.5 mM), DDC (2 mM), IMZ (1 mM), ABA (0.5 mM), DMTU (10 mM), and DPI (0.1 mM), were prepared in accordance with published literature (*Peng & Harberd, 2002*; *Sagi & Fluhr, 2006*; *Deng et al., 2010*; *Huang et al., 2016*; *Lu et al., 2019*; *Samuilov et al., 2021*) and our own preliminary experiments.

### Seed germination and treatment

The tested variety is the main wheat variety cultivated in Northeast China, "Longmai 35" (selected by the Heilongjiang Academy of Agricultural Sciences). Evenly sized and plump wheat seeds were selected, which were surface-sterilized for 5 min using 0.1% HgCl$_2$, followed by a 2 h soak in water. Finally, the seeds were sown on double-layer filter-paper (No. 1, Whatman®, Clifton, NJ, USA) under different NaCl concentrations. Dark cultivation was conducted in a culture box (KBW 400; BINDER®, Tuttlingen, Germany) set at different temperatures, while maintaining a relative humidity of 60% (*Iqbal, Ashraf & Jamil, 2006*). Wheat seed germination was observed and counted after germination for

24, 48, and 72 h (*Pessarakli, Tucker & Nakabayashi, 1991*). Germination was defined as a root length of 0.5 mm or more.

In this experiment, the wheat seeds were divided into eight groups (Fig. 1). The "combined stress" group refers to the seeds treated with 50 mM NaCl and 30 °C simultaneously.

In group 1, four levels of heat stress (20 °C, 25 °C, 30 °C and 35 °C) were applied to the wheat seeds under salt-free conditions.

In group 2, wheat seeds were treated with four concentrations of NaCl (0, 50, 100, and 200 mM) at 20 °C.

In group 3, four levels of heat stress (20 °C, 25 °C, 30 °C and 35 °C) were applied to the wheat seeds under low salt (50 mM Nacl) stress.

In group 4, four concentrations (0, 0.5, 2, and 10.0 mM) of GABA were applied to wheat seeds at 20 °C.

In group 5, four concentrations (0, 0.5, 2, and 10 mM) of GABA were applied to wheat seeds under combined stress.

In group 6, 2 mM GABA was applied to wheat seeds under combined stress, in conjunction with 10 mM DMTU (an ROS scavenger), 0.1 mM DPI (an NOX inhibitor), or 1 mM IMZ (an NOX inhibitor).

In group 7, 2 mM GABA was applied to wheat seeds under combined stress, in conjunction with 10 mM $H_2O_2$, 2 mM DDC, (an inhibitor of SOD), or 2 mM ATZ (an inhibitor of CAT).

In group 8, 2 Mm GABA was applied to wheat seeds under combined stress, in conjunction with 0.1 mM Flu (an inhibitor of ABA), 0.5 mM ABA, or 0.5 mM GA.

In the tests, a 1-mL spray bottle was used for each treatment, and the solution (*i.e.*, GABA, DMTU, DPI, IMZ, $H_2O_2$, DDC, ATZ, FLU, ABA, or GA) was evenly sprayed into the culture dish. Every test was performed at least five times, and 100 seeds were used in each replicate for the germination test. The proportion of normally germinated seeds among all tested seeds, following incubation for a certain period, was used to compute the germination rate (GR).

## Assay of $H_2O_2$, ABA, and GABA contents

Hydrogen peroxide ($H_2O_2$) content was quantified using the xylenol orange assay (*Gay, Collins & Gebicki, 1999*; *Zhang, Deng & Li, 2018*). This assay relies on the oxidation of Fe(II) by peroxide, followed by colorimetric detection of Fe(III) complexed with the sodium salt of xylenol orange. To perform the assay, 1 mL of reagent solution (25 mM $FeSO_4$ and 25 mM $(NH_4)_2SO_4$ in 2.5 M $H_2SO_4$) was added to 100 mL of 125 µM xylenol orange and 100 mM sorbitol. After centrifuging ground eggplant leaves at 5,000 g for 10 min, 100 µL of supernatant was mixed with 1 mL of xylenol orange reagent. Following a 30-min incubation period, absorbance of the Fe(III)–xylenol orange complex was measured at 560 nm.

The ABA and GABA contents in germinated wheat seeds and sprouts were determined using high-performance liquid chromatography (HPLC).

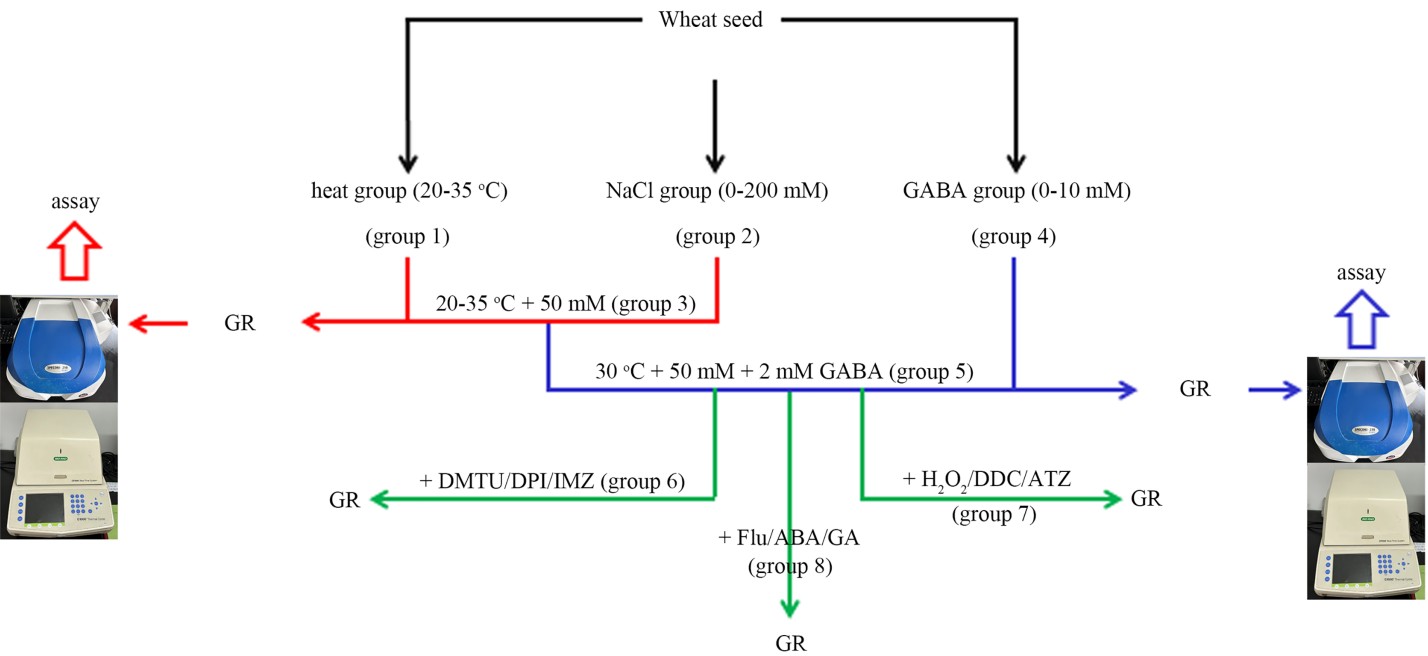

**Figure 1 A simple flow chart for the experiment design.** The experimental flowchart shows the detection of the effects of different concentrations of NaCl and GABA, as well as different temperatures on germination rate (GR) of wheat seed. The effects of GABA, combined with various reagents, on wheat seed germination under combined stress of 30 °C and 50 mM NaCl were also investigated. GABA, γ-aminobutyric acid; ABA, abscisic acid; GA, gibberellin; DPI, diphenyleneiodonium chloride; DMTU, dimethylthiourea; IMZ, imidazole; DDC, diethyldithiocarbamic acid; ATZ, amino-triazole.                                

## Enzyme activity assays

For GAD activity measurement (*Bai et al., 2009*), 0.1 M potassium phosphate buffer (pH 5.8) containing 2 mM β-mercaptoethanol, 2 mM EDTA, and 0.2 mM 5-pyridoxal phosphate was used as the extraction solution. Samples (1 g) were homogenized with 5 mL of extract on ice and centrifuged at 15,000 g for 15 min. The resulting crude enzyme solution was mixed with 200 μL of substrate (1% Glu, pH 5.8), incubated at 40 °C for 2 h, and terminated by heating at 90 °C for 5 min. GABA content in the filtrate was detected using a 0.45-μm membrane filter, with enzyme activity defined as the amount of GABA produced per hour at 40 °C.

NOX activity was evaluated using a Plant NADPH oxidase ELISA Kit (GMS50096.3 v.A; GenMed Scientific, Plymouth, MN, USA) following the manufacturer's instructions, and absorbance was measured at 340 nm.

SOD activity in wheat sprouts was determined using the method of *Dhindsa, Plumb-Dhindsa & Thorpe (1981)*. Enzyme extract was mixed with 0.1 M phosphate buffer (pH 7.8), and absorbance at 560 nm was recorded after 15 min of light exposure. One unit (U) of SOD activity was defined as the enzyme amount inhibiting 50% NBT photoreduction.

CAT activity was assessed following the method of *Zhang & Kirkham (1994)*. Reaction buffer (3 mL) was combined with 50 μL enzyme extract from wheat sprouts, and the reaction was initiated by adding 15 mM $H_2O_2$. Activity was measured by monitoring $H_2O_2$ consumption at 240 nm for 3 min (E = 39.4 $mM^{-1}$ $cm^{-1}$).
The soluble protein content was determined using the method of *Bradford (1976)*, using bovine serum albumin as a standard.

## Quantitative real-time PCR (qRT–PCR) for antioxidant enzyme encoding genes

Wheat seeds were treated under various conditions: normal cultivation (0 mM NaCl + 20 °C), salt stress (50 mM NaCl + 20 °C), high temperature stress (0 mM NaCl + 30 °C), combined salt and high temperature stress (50 mM NaCl + 30 °C), and combined salt, high temperature stress with exogenous GABA (50 mM NaCl + 30 °C + 2 mM GABA) for 8 and 16 h. Each treatment was replicated three times biologically, implying that three separate groups of wheat seeds were collected under each condition.

Total RNA was extracted from germinated wheat seeds and sprouts using the Trizol protocol (Thermo Fisher Scientific, Waltham, MA, USA), which involved a series of steps including extraction, purification, homogenization, separation, precipitation, washing, dissolution, and determination of concentration and purity. This protocol was consistently applied to extract RNA from all samples, and the purified RNA was subsequently stored at −80 °C. In each treatment, the mortar and pestle were sterilized at high temperature, pre-cooled with liquid nitrogen, and 100 mg of seed tissue was ground into a fine powder. The experiment was subsequently carried out on ice. The ground tissue was then transferred to 1.5 mL Eppendorf tubes, and total RNA was extracted using Trizol reagent (Invitrogen, Waltham, MA, USA). RNA integrity and concentration were assessed through 1% denatured agarose gel electrophoresis, while RNA quality was evaluated using Nanodrop OneC (Thermo Fisher Scientific, Waltham, MA, USA). cDNA was synthesized using the ReverTra AceTM qPCR RT Master Mix with gDNA Removal kit (TOYOBO Co., Osaka, Japan). Specific primers for SOD and CAT were designed using the Primer-BLAST online tool (http://www.ncbi.nlm.nih.gov/tools/primer-blast/). The sequences of qRT-PCR primers are listed in Table 1. Using Bio Rad Laboratories Inc., Hercules, CA, USA 2.423 instruments, CFX96 TouchTM real-time PCR detection system (Bio Rad, Hercules, CA, USA) and SYBR Premix Ex Taq II Kit (Takara, Dalian, China), the expression patterns of two antioxidant enzymes were analyzed using real-time quantitative polymerase chain reaction (qRT PCR). Each treatment was repeated three times. cDNA from wheat seeds under normal cultivation conditions was used as a template to create standard curves for target genes and reference genes. Finally, the amplification efficiency was calculated through slope analysis, and the $2^{-\Delta\Delta Ct}$ was used for calculation (*Livak & Schmittgen, 2001*; *Ogonowska & Nakonieczna, 2020*). Target gene data were standardized using *actin* as a reference gene (*Zou et al., 2018*; *Cai et al., 2011*).

## Data analysis

SPSS 13.0 (SPSS Inc., Chicago, IL, USA) was used for data analysis. The results are shown as the mean ± standard deviation. According to the Duncan's multiple range test, means accompanied by the same letter indicate no significant difference between them ($p < 0.05$).

**Table 1 Primer sequence list.**

| S/NO | Gene name | Gene ID | Amplifier length (bp) | Forward | Reverse |
|---|---|---|---|---|---|
| 1 | Actin | XM_044554036.1 | 176 | CCTTCGTTTGGACCTTGCTG | AGCTGCTCCTAGCCGTTTCC |
| 2 | MnSOD | XM_044478966.1 | 156 | GAACCTCAAGCCCATCAGCG | AAAGCTAGCCACACCCATCC |
| 3 | CAT | NM_001405704.1 | 103 | CCATGAGATCAAGGCCATCT | ATCTTACATGCTCGGCTTGG |

## RESULTS

### Effects of heat stress, salinity, and GABA on germination rate

Considering that the GR of wheat seeds stabilizes after 72 h of germination, this analysis focuses on the final GR of wheat seeds after 72 h, examining the impacts of heat stress, salinity, and GABA (Figs. 2 and 3).

The final GRs of wheat seeds at four different temperatures (20 °C, 25 °C, 30 °C, and 35 °C) showed an initial increase followed by a decrease, with GR values of 88%, 98%, 63%, and 17%, respectively (Fig. 2A; $p < 0.05$). Similarly, at 20 °C, the GR for NaCl concentrations of 0, 50, 100, and 200 mM followed the same trend, with GR values of 89.6%, 96.6%, 63.2%, and 15.4%, respectively (Fig. 2B; $p < 0.05$). Under low salinity conditions (50 mM NaCl), increasing the temperature from 20 °C to 35 °C resulted in a sharp decline in GR, with values of 95.6%, 79.8%, 21.2%, and 3.4%, respectively (Fig. 2C; $p < 0.05$). At 20 °C, GABA treatment showed a slight increase in GR as the concentration increased from 0 to 10 mM; for instance, a 2-mM GABA treatment resulted in a GR of 5.5% higher than that of the water control (Fig. 2D).

Under combined stress (50 mM NaCl + 30 °C), the final GR of wheat seeds treated with 0, 0.5, 2, and 10 mM GABA increased significantly, reaching 21.2%, 38.4%, 48.4% and 58.4%, respectively (Fig. 3A; $p < 0.05$). Compared to the combined stress + GABA treatment group, additional treatment with DMTU, DPI, and IMZ decreased GR by 27.6%, 19.7%, and 17.6%, respectively (Fig. 3B; $p < 0.05$). However, further treatment with $H_2O_2$, ATZ, and DDC increased GR by 14.6%, 25.1%, and 42.3%, respectively (Fig. 3C; $p < 0.05$). Additionally, further treatment with ABA, Flu, and GA decreased GR by 27.0%, and increased it by 14.5% and 19.1%, respectively (Fig. 3D; $p < 0.05$).

### Effects of combined stress and GABA on the contents of $H_2O_2$, ABA, GABA and GAD activities of GAD and NOX

The impacts of combined stress conditions (30 °C + 50 mM NaCl) and combined stress GABA treatment on $H_2O_2$, ABA, and GABA contents, as well as GAD and NOX enzyme activities, were assessed within the first 24 h after germination (Fig. 4).

Compared to the control group (20 °C + 0 mM NaCl), combined stress significantly reduced $H_2O_2$ and GABA contents and the activities of GAD and NOX enzymes, while increasing ABA accumulation. For example, after 2, 6, 10, 14, 18, and 22 h of treatment, the $H_2O_2$ content in wheat seeds and sprouts in the combined stress group decreased by 60.2%, 51.8%, 40.0%, 30.3%, 35.5%, and 32.1%, respectively, compared to those of the control group (Fig. 4A; $p < 0.05$).

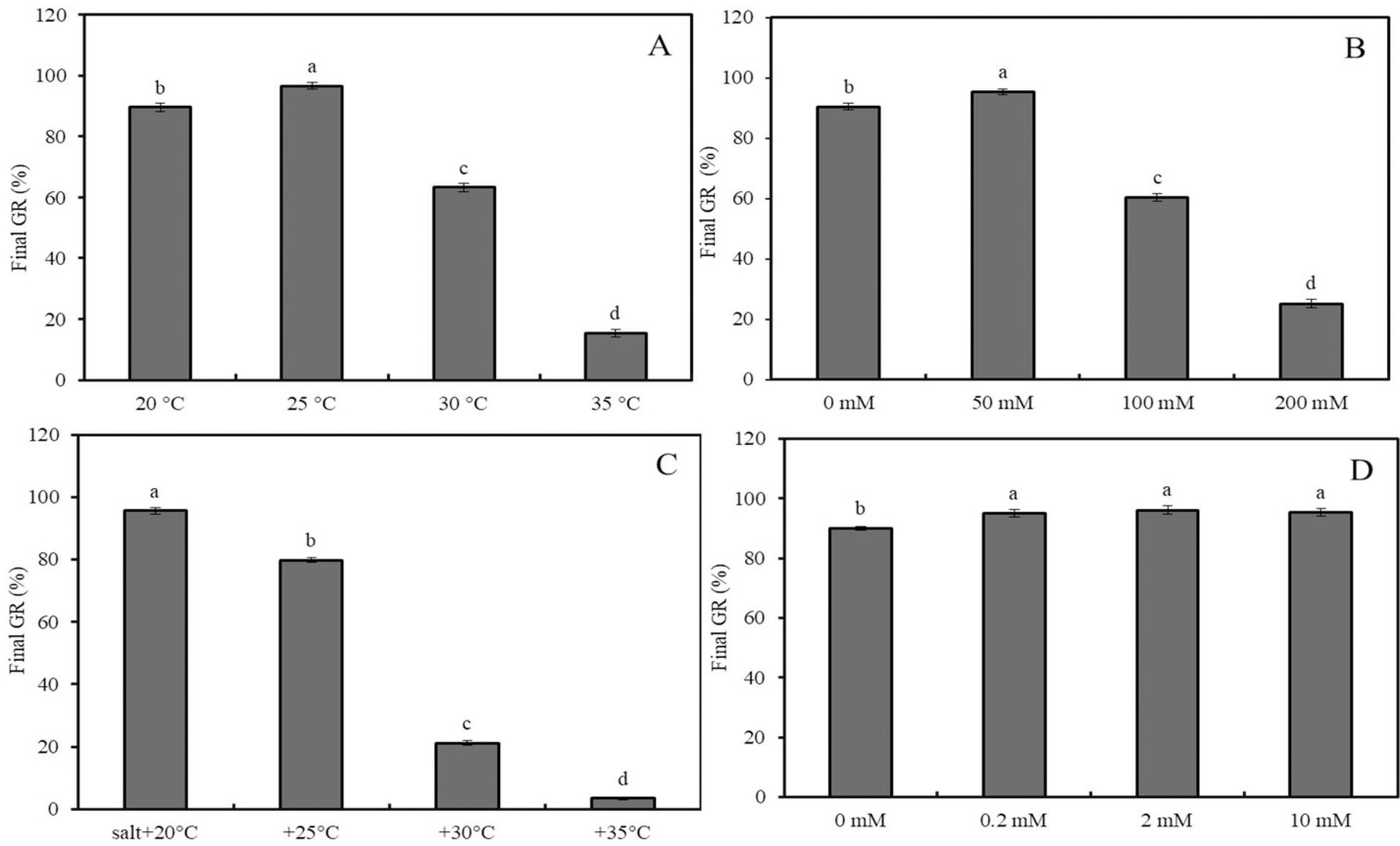

**Figure 2  Effects of salinity and heat stress on seed germination.** Effects of heat stress (A), salinity (B), combined stress of salinity and heat stress (C), and GABA (D) on final germination rate of wheat seeds. Bars represent standard deviation of the mean (*n* = 3); means associated with the same letter are not significantly different (*p* < 0.05). GABA, γ-aminobutyric acid. 

Conversely, ABA content in wheat seeds after 2, 6, 10, 14, 18, and 22 h of combined stress were 61.5%, 95.5%, 141.5%, 204.1%, 269.5%, and 336.6% higher than those in the control group, respectively (Fig. 4B; *p* < 0.05). Additionally, after 22 h of treatment, endogenous GABA content and the activities of GAD and NOX in wheat seeds and sprouts in the combined stress group were 71.9%, 84.1%, and 84.7% lower than those in the control group, respectively (Figs. 4C–4E; *p* < 0.05).

In wheat seeds subjected to combined stress +GABA treatment, there was an increase in $H_2O_2$ and GABA contents, GAD and NOX enzyme activities, and a decrease in ABA content compared to seeds treated with combined stress alone (Fig. 4). Specifically, following 2 h of combined stress + GABA treatment, the levels of $H_2O_2$, ABA, GABA content, as well as GAD and NOX enzyme activities in wheat seeds were significantly higher by 104.0%, −24.5%, 53.1%, 145.6%, and 237.9%, respectively, compared to seeds treated with combined stress alone (Fig. 4; *p* < 0.05). Similarly, after 22 h of germination, these values were elevated by 31.9%, −48.5%, 114.9%, 406.1%, and 201.8%, respectively (Fig. 4; *p* < 0.05).

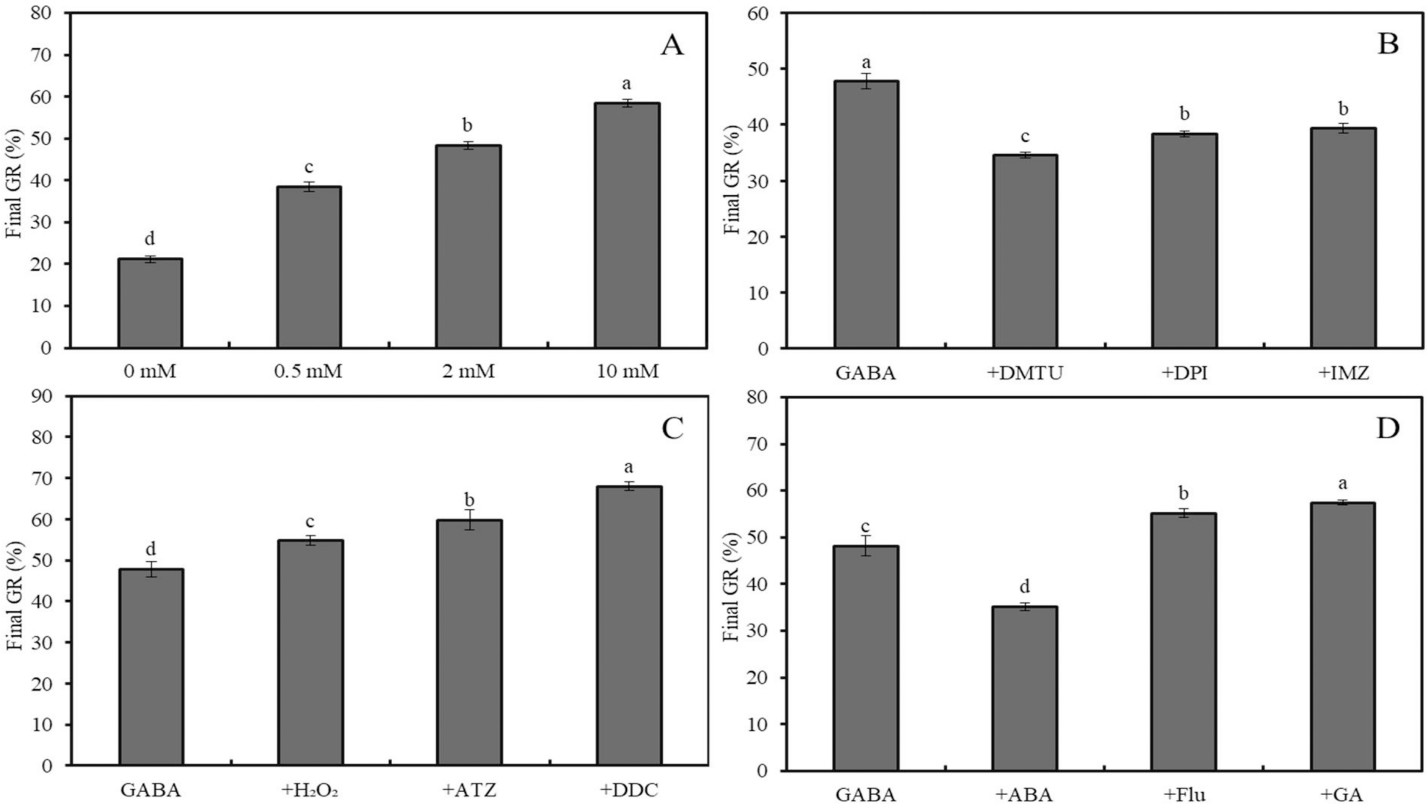

**Figure 3 Effects of GABA on seed germination under combined stress.** Effects of 0–10 mM GABA (A), 2 mM GABA + antioxidant (10 mM DMTU, 0.1 mM DPI or 1 mM IMZ) (B), 2 mM GABA + oxidant (10 mM $H_2O_2$, 2 mM ATZ or 2 mM DDC) (C), and 2 mM GABA + hormone (0.5 mM ABA, 0.1 mM Flu or 0.5 mM GA) (D) on final GR of wheat seeds under combined stress (30 °C + 50 mM NaCl). Bars represent standard deviation of the mean ($n$ = 3); means associated with the same letter are not significantly different ($p < 0.05$). ABA, abscisic acid; GA, gibberellin; GABA, γ-aminobutyric acid; DPI, diphenyleneiodonium chloride; DMTU, dimethylthiourea; IMZ, imidazole; DDC, diethyldithiocarbamic acid; ATZ, aminotriazole; Flu, fluridone.

## Effects of combined stress and GABA on activities of SOD and CAT activities and *MnSOD* and *CAT* expression levels

The effects of temperature, salinity, combined temperature and salinity, and combined temperature, salinity, and GABA on SOD and CAT activities, as well as *MnSOD* and *CAT* genes expressions, in germinated seeds and sprouts were investigated (Fig. 5). High temperature (30 °C) led to a 39.0%, 36.0%, 65.4%, and 57.9% decrease in SOD and CAT enzyme activities and *MnSOD* and *CAT* gene expressions, respectively, in wheat seeds after 8 h of germination compared to seeds at 20 °C (Fig. 5; $p < 0.05$). In contrast, mild salt stress (50 mM NaCl) activated SOD and CAT enzyme activities and increased the expression levels of *MnSOD* and *CAT* compared to those of the control at 20 °C. After treatment with 50 mM NaCl for 8 h, SOD and CAT enzyme activities, as well as *MnSOD* and *CAT* gene expression levels in wheat sprouts at 20 °C, were 16.3%, 17.2%, 14.0%, and 31.3% higher than those in the water-treated control group (Fig. 5; $p < 0.05$).

Under combined stress, after 8 h of germination, SOD and CAT activities were 70.4% and 71.4% lower in the seeds compared to those of seeds subjected to the same salt concentration (50 mM NaCl) at 20 °C (Figs. 5A, 5B; $p < 0.05$). Similarly, the expression

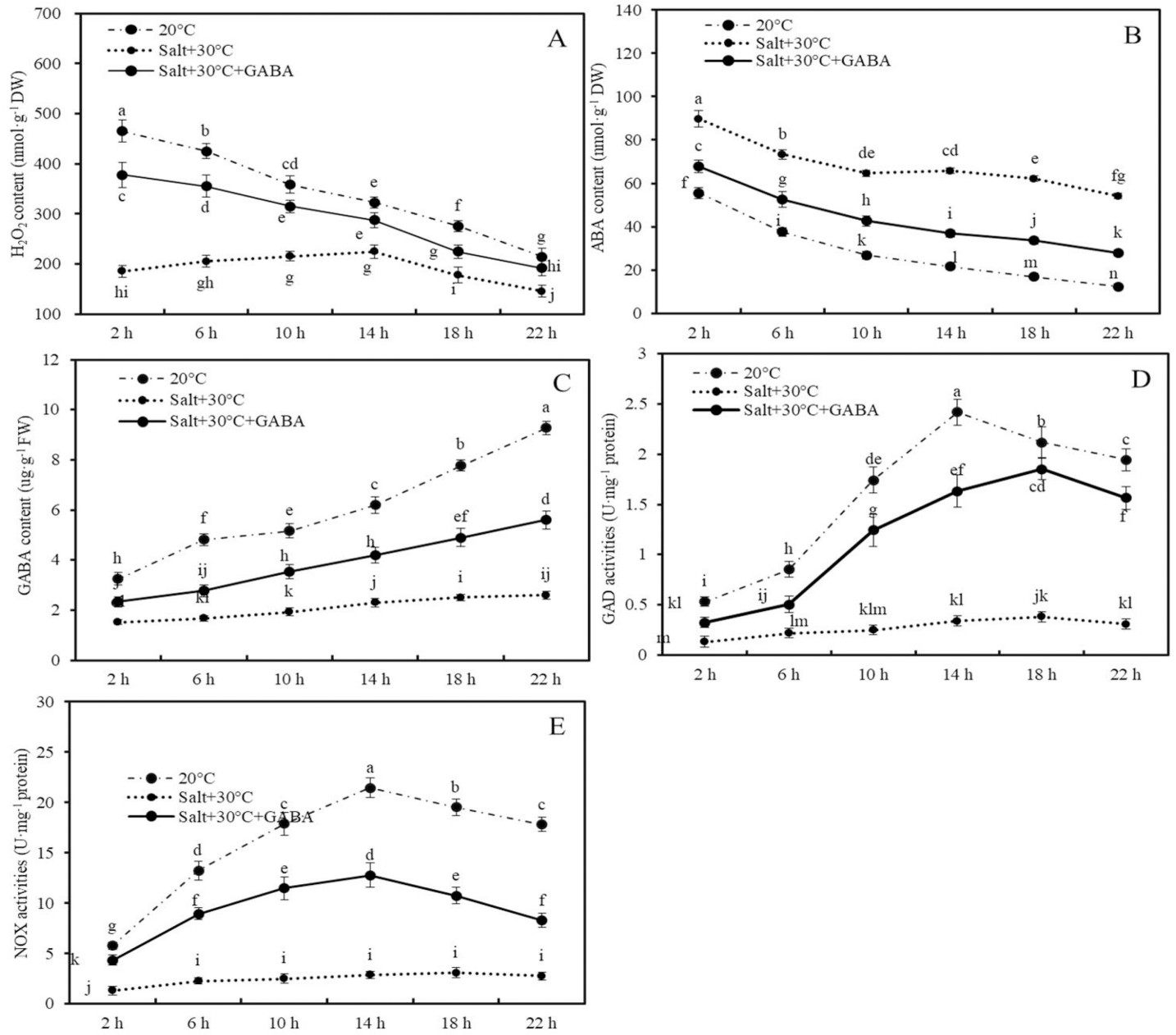

**Figure 4 Contents of H₂O₂, ABA and GABA, and activities of GAD and NOX.** Effects of GABA and combined stress (30 °C + 50 mM NaCl) on $H_2O_2$ content (A), ABA content (B), GABA content (C), GAD activities (D), and NOX activities (E) in germinated wheat seeds and sprouts within the first 22 h. Bars represent standard deviation of the mean ($n = 3$); means associated with the same letter are not significantly different ($p < 0.05$). GABA, γ-aminobutyric acid; ABA, abscisic acid; GAD, glutamic decarboxylase.

levels of *MnSOD* and *CAT* under combined stress were 81.1% and 88.6% lower than those under salt stress at 20 °C, respectively (Figs. 5C, 5D; $p < 0.05$).

Furthermore, the effects of GABA on SOD and CAT activities and *MnSOD* and *CAT* gene expressions in wheat seeds after 8 and 16 h of germination under combined stress (30 °C + 50 mM NaCl) were examined (Fig. 5). Combined stress + GABA treatment resulted in a 155.2%, 229.2%, 153.5%, and 181.2% increase in SOD and CAT enzyme

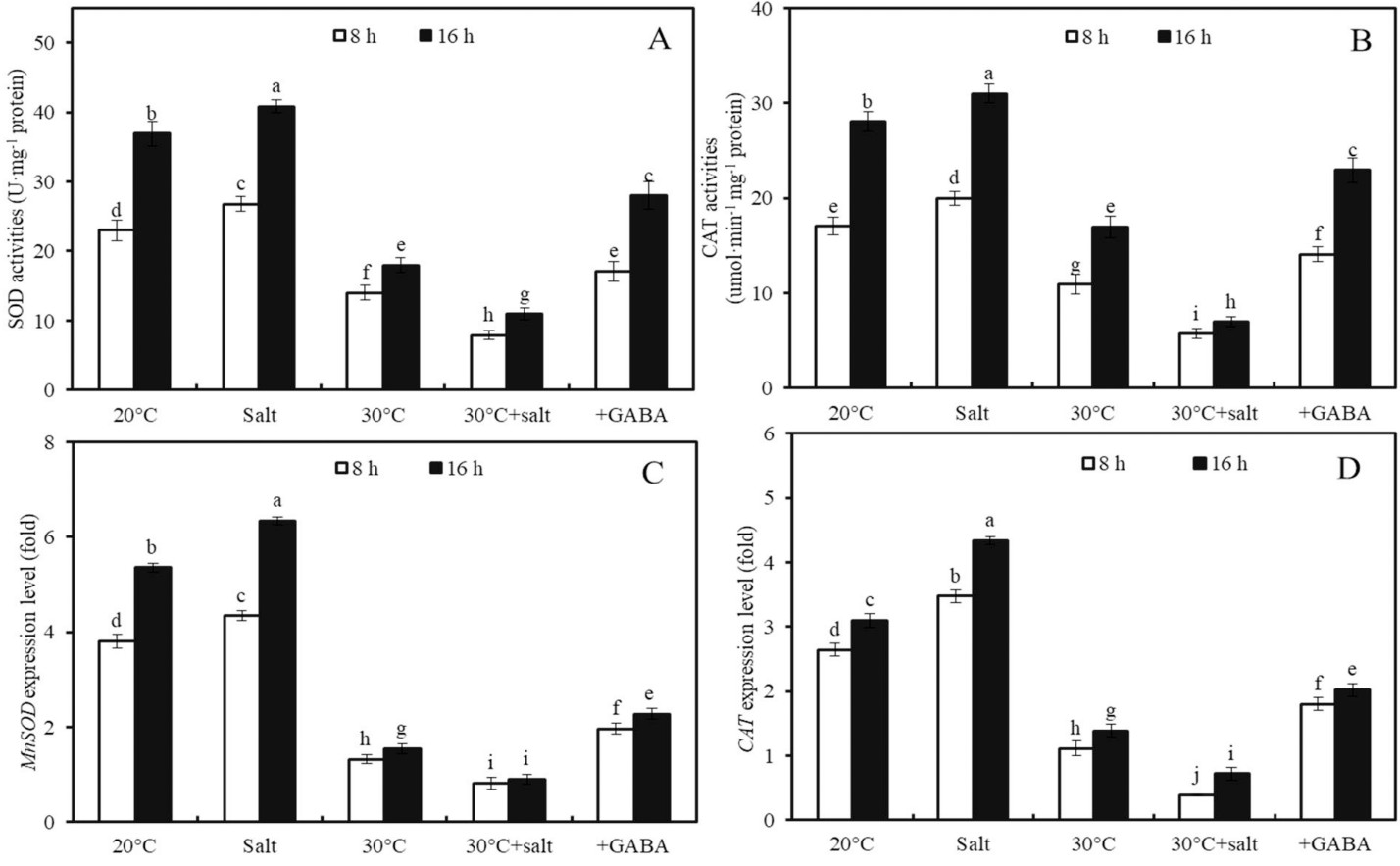

**Figure 5 Antioxidant enzyme activities and gene expression levels.** Effects of temperature and salinity (50 mM NaCl) alone and in combination on activities of SOD (A) and CAT (B), and expression levels of *MnSOD* (C) and *CAT* (D) in germinated wheat seeds and sprouts within the first 22 h. Bars represent standard deviation of the mean ($n$ = 3); means associated with the same letter are not significantly different ($p < 0.05$). GABA, γ-aminobutyric acid; SOD, superoxide dismutase; CAT, catalase.               

activities and MnSOD and CAT gene expression levels, respectively, in wheat seeds at 16 h after germination compared to seeds treated with combined stress alone (Fig. 5; $p < 0.05$).

## DISCUSSION

Our research found that while severe salt stress decreases GR, a low concentration of NaCl (50 mM) can increase the GR of wheat, potentially due to mild salt stress inducing ROS proliferation and promoting seed dormancy breaking (*Bailly, El-Maarouf-Bouteau & Corbineau, 2008*; *Bailly, 2019*). Similarly, although high temperatures inhibit seed germination, moderate heat stress (such as 25 °C) can more effectively promote dormancy breaking and increase GR compared to the optimal germination temperature of 20 °C (*Khaeim et al., 2022*). Interestingly, under low salt stress (50 mM), a slight temperature increase to 25 °C significantly decreases GR compared to the high GR at 20 °C. However, further temperature increases to moderate heat stress (30 °C) severely inhibit wheat germination. This indicates that the combined stress experienced during seed germination

is not simply additive, which aligns with the notion that certain combined stresses can cause more significant damage to plants (*Mittler, 2006*).

In subsequent studies, "combined stress" refers to placing wheat seeds under low salt stress (50 mM NaCl) and moderate heat stress (30 °C) simultaneously, and observing the effects of GABA on their germination. While GABA has a slight promoting effect on germination at the optimal temperature (20 °C), it significantly increase the GR of wheat seeds under combined stress.

Given the crucial role of ROS in seed germination (*Bailly, El-Maarouf-Bouteau & Corbineau, 2008*; *Bailly, 2019*). ROS-specific scavenger DMTU and two specific inhibitors of NOX (a major ROS-producing enzyme), DPI and IMZ (*Sagi & Fluhr, 2006*), were used alongside GABA to treat wheat seeds under combined stress. The results showed that the promoting effects of GABA on wheat germination under combined stress could be attenuated by DMTU (a well-known ROS scavenger; *Liu et al., 2023*), as well as DPI and IMZ (NOX specific inhibitors; *Ishibashi et al., 2010*; *Zhang, Deng & Li, 2018*). Meanwhile, the promoting effect of GABA can be enhanced by $H_2O_2$, and antioxidant enzyme inhibitors DDC, (SOD specific inhibitor; *Samuilov et al., 2021*) and ATZ (CAT enzyme specific inhibitor; *Deng et al., 2010*). This suggests that the enhanced germination of wheat seeds under combined stress by GABA is closely related to ROS accumulation levels, indicating that combined stress causes a severe ROS deficiency, hindering wheat seeds from breaking dormancy and entering the germination stage.

Additionally, the study explored GABA's effect on wheat seed germination under combined stress at the hormone level. It was found that exogenous ABA weakened GABA's promoting effect, while GA (an ABA antagonist; *Peng & Harberd, 2002*) and Flu (an ABA inhibitor; *Lu et al., 2019*) enhanced it. This suggests that the reduced GR caused by combined stress is related to higher ABA accumulation, and GABA may accelerate ABA degradation.

Further research detected the contents of $H_2O_2$, ABA, and GABA as well as GAD enzyme activity, within 24 h after the initiation of wheat seed germination under combined stress. The results showed a significant decrease in $H_2O_2$ content and a significant increase in ABA accumulation compared to those of the control group (20 °C + 0 mM NaCl). This indicates that combined stress reduces $H_2O_2$ content while increasing ABA accumulation, preventing seeds from effectively breaking dormancy and entering the germination stage, thereby decreasing GR. Combined stress also significantly decreased GABA content and GAD enzyme activity in wheat seeds, suggesting that endogenous GABA deficiency is a key factor preventing germination.

Previous studies have shown that bean seed germination leads to increased GABA (*Kuo et al., 2004*), and salt stress promotes GABA accumulation during wheat seed germination (*Al-Quraan, Al-Ajlouni & Obedat, 2019*). This indicates that increased GABA content promotes seed germination under both normal and adverse conditions, and combined stress severely inhibits wheat seed germination, resulting in a much lower GABA content than those at normal germination levels. This insufficient GABA may be a crucial factor preventing germination under combined stress.

Further study showed that GABA treatment not only significantly increased $H_2O_2$ content and decreased ABA content in wheat seeds under combined stress, but also increased GABA accumulation and GAD enzyme activity. This suggests that exogenous GABA supplementation activates GAD enzyme activity to generate more endogenous GABA, and also increases $H_2O_2$ content in wheat seeds, ultimately leading to ABA degradation, releasing dormancy, and entering the germination stage. This raises the question: why does exogenous GABA cause an increase in $H_2O_2$ accumulation in germinated seeds and sprouts?

Previous studies have shown that treating postharvest apples with GABA significantly increases $H_2O_2$ content and the activities of NOX and antioxidant enzymes such as SOD and CAT (*Zhu et al., 2022*). These enzymes play a crucial role in seed germination, particularly under challenging conditions (*Wang et al., 2009*), *MnSOD* and *CAT* are key genes encoding SOD and CAT enzymes in wheat plants mitochondria (*Baek & Skinner, 2003*). Therefore, in our research, we measured the activities of NOX, SOD and CAT enzymes, as well as the expression levels of MnSOD and CAT genes.

Our results indicated that the activities of NOX, SOD and CAT enzymes, along with the expression levels of MnSOD and CAT genes, were inhibited by combined stress but could be restored by GABA treatment. This raises the question: why does GABA activate NOX and antioxidant enzyme activities (SOD and CAT) in wheat seeds under combined stress?

We speculate that GABA, a non protein amino acid, may play a signaling role as its content increases during seed germination (*Kuo et al., 2004*). In the present study, exogenous GABA supplementation might stimulate NOX enzyme activity, leading to ROS proliferation (*Zhu et al., 2022*). This ROS proliferation can activate the antioxidant system (*Deng et al., 2023*), resulting in the synthesis of enzymes such as SOD and CAT and enhanced gene expression (*Mylona, Polidoros & Scandalios, 2007*).

The upregulation of these key antioxidant enzyme genes, especially *MnSOD* (*Cheng et al., 2016*), suggests that GABA activates NOX to produce more superoxide anions, which are dismutated by SOD to generate $H_2O_2$. This process helps seeds degrade ABA and break dormancy (*Liu et al., 2010*; *Cheng et al., 2022*) and promotes seed metabolism, such as starch degradation (*Cheng et al., 2018*; *Sheng et al., 2018*), allowing for the synthesis of more antioxidant enzymes to cope with ROS toxicity induced by combined stress during germination. The activated antioxidant system also prevents excessive ROS from inhibiting germination (*Bailly, El-Maarouf-Bouteau & Corbineau, 2008*; *Bailly, 2019*).

For a better understanding, we propose a hypothetical model (Fig. 6): combined stress induces ABA accumulation, preventing wheat seeds from breaking dormancy. GABA activates NOX to produce more ROS, accelerating ABA degradation and promoting seed germination. The ROS signal mediated by NOX can activate the synthesis of related antioxidant enzymes (*Deng et al., 2023*), preventing excessive ROS accumulation and inhibiting germination. Additionally, germinating seeds exhibit more vigorous metabolism compared to dormant seeds, which may explain their higher enzyme activity and gene expression levels.

This work provides valuable theoretical and practical insights for the germination and growth of crop seeds in challenging and changing environments, offering new methods

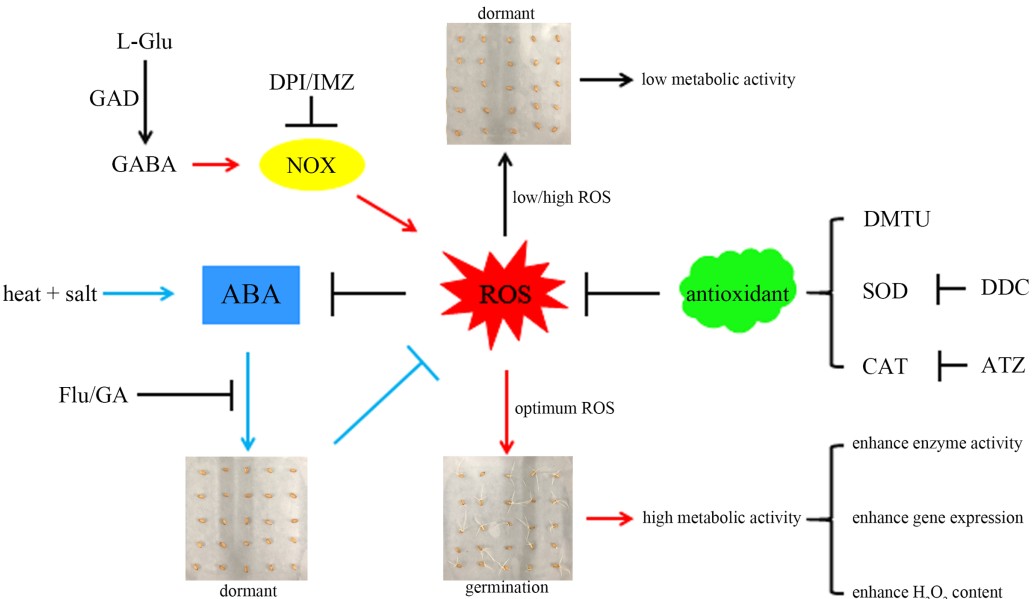

**Figure 6 A hypothetical model.** A hypothetical model based on GABA activation of NOX to generate more ROS and relieve the inhibition of ABA on seed germination is proposed here. Here, the combined stress of high temperature and salinity induces the accumulation of ABA and leads to insufficient ROS, thereby inhibiting seed germination. GABA can activate NOX, which can generate more ROS, thereby relieving the inhibitory effect of ABA on seed germination. The sharp and blunt arrows indicate the positive and negative effects, respectively. For details, please refer to the text. GABA, γ-aminobutyric acid; GAD, glutamate decarboxylase; NOX, NADPH oxidase; SOD, superoxide dismutase; CAT, catalase; ROS, reactive oxygen species; ABA, abscisic acid; GA, gibberellin; DPI, diphenyleneiodonium chloride; DMTU, dimethylthiourea; IMZ, imidazole; DDC, diethyldithiocarbamic acid; ATZ, aminotriazole; Flu, fluridone.                               

and inspiration for sustainable agriculture (*Miransari & Smith, 2019*). However, the relationship between the ROS signal mediated by GABA-activated NOX and other physiological changes during seed germination (such as the acceleration of starch degradation by GABA-activated α-amylase; *Cheng et al., 2018*) remains unclear. Future research could utilize transcriptomics combined with metabolomics to further analyse these regulatory mechanisms.

## CONCLUSIONS

In the present study, several notable conclusions can be drawn. Initially, it was observed that mild salt stress (*e.g.*, 50 mM NaCl) and mild heat stress (*e.g.*, 25 °C) both enhance the GR of wheat seeds. However, when mild salt stress is combined with mild heat stress or higher, there is a significant decrease in GR. Additionally, the application of exogenous GABA was found to greatly improve seed germination under combined stress conditions. This beneficial effect can be further amplified by the presence of $H_2O_2$, flu, and antioxidant enzyme inhibitors, while it may be diminished by DMTU, ABA, and NOX inhibitors. Furthermore, the results indicate that combined stress leads to an increase in ABA content and a decrease in $H_2O_2$ and GABA contents, as well as a reduction in the activities of GAD, NOX, SOD, and CAT enzymes, along with a decrease in antioxidant enzyme gene expression. However, the application of GABA was able to reverse these effects.

Additionally, it was observed that GABA may stimulate NOX activity, leading to an increase in $H_2O_2$ production, which in turn accelerates the degradation of ABA induced by combined stress, ultimately enhancing seed germination. These findings suggest that utilizing GABA treatment for crop seeds could effectively improve germination rates under combined stress conditions, providing a potential strategy for enhancing agricultural productivity in challenging environments.

## ACKNOWLEDGEMENTS

We appreciate the experimental instrument and equipment support provided by the National Coarse Cereals Engineering Research Center of China.

### Funding

This work was financially supported by the National Key Research and Development Program of China (Grant number: 2018YFD1000704), the Postdoctoral Science Foundation Funded General Project of Heilongjiang Province (Grant number: LBH-Z19195), and the Scientific Research Project for People Returned after Further Learning and Talent Introduction of Heilongjiang Bayi Agricultural University (Grant number: XYB2014-02). The funders had no role in study design, data collection and analysis, decision to publish, or preparation of the manuscript.

### Grant Disclosures

The following grant information was disclosed by the authors:
National Key Research and Development Program of China: 2018YFD1000704.
Heilongjiang Province: LBH-Z19195.
Heilongjiang Bayi Agricultural University: XYB2014-02.

### Competing Interests

The authors declare that they have no competing interests.

### Author Contributions

- Song Yu conceived and designed the experiments, performed the experiments, analyzed the data, prepared figures and/or tables, authored or reviewed drafts of the article, and approved the final draft.
- Zhihan Lian performed the experiments, analyzed the data, prepared figures and/or tables, authored or reviewed drafts of the article, and approved the final draft.
- Lihe Yu conceived and designed the experiments, authored or reviewed drafts of the article, and approved the final draft.
- Wei Guo performed the experiments, analyzed the data, prepared figures and/or tables, and approved the final draft.
- Chunyu Zhang performed the experiments, analyzed the data, prepared figures and/or tables, and approved the final draft.

- Yifei Zhang conceived and designed the experiments, analyzed the data, prepared figures and/or tables, authored or reviewed drafts of the article, and approved the final draft.

## Data Availability

The raw measurements are available in the Supplemental Files.

## Supplemental Information

Supplemental information for this article can be found online at http://dx.doi.org/10.7717/peerj.17907#supplemental-information.

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
