# Peer review of "Gamma-aminobutyric acid elicits H2O2 signalling and promotes wheat seed germination under combined salt and heat stress"

_PeerJ, doi:10.7717/peerj.17907_

## Round 0.1 · original submission · Major Revisions

Abiotic stress is an important phenomenon we should find how some row crops like wheat respond to this adverse environmental condition and how we reduce the adverse impact of it. Your research encompasses valuable insights significant for understanding the impact of Gamma-aminobutyric acid on wheat seed germination under abiotic stress conditions. Nevertheless, it is imperative to address certain technical intricacies in refining your article. I strongly advocate for a comprehensive review of the reviewers' suggestions, coupled with a discerning consideration of each recommendation. In cases where you may disagree with specific suggestions, providing clear and well-justified rationale for your perspective would prove beneficial.

·

Basic reporting

This manuscript described the use of GABA to promote seed germination during abiotic stress conditions. The authors analyzed several abiotic stress combinations and exogenous chemical treatments to understand the seed germination rate. The author showed how mixed abiotic stress and exogenous treatment can enhance several components and alter the different pathways. They also showed the marker gene expression pattern during different combinations of stress. The overall study is important, but some lacking points need further improvement.
1. This study is important to understand the impact of salt concentration along with temperature stress, but how the crosstalk of genes and these pathways governing during this stress is lacking in explanation. I will encourage you to provide the valid crosstalk of pathways involved in these components that are enhanced during exogenous GABA treatment.
2. It seems that several combinations of stress along with so many chemical treatments make confusion and clear elaboration of each work not justified, it is really important to understand the pathways of each exogenous treatment and their role. I will recommend summarizing the results of each treatment and their effects in tabular form, which will allow the readers to understand better. Also try to describe the involved pathways or their mode of action latest in the discussion part.
3. The figure legends are not written well, and need to be described more with all the small details such as what statistics were applied to justify the significance of each graph. So many errors in English writing in legends, try to check all and correct it.
4. In Figure 2 legends, I can see that figure 2a has 0.5, 2 and 10mM GABA concentration, but not described in the legend. Check all legends and describe them.
5. Abstract must be precise and clear, explanation of techniques used, and methodology is not required in abstract. Make the abstract clear and precise.
6. Lines 205-206 need to be rewritten; something is missing in the sentence.
7. Line 215, if during writing primer list came first, so it will be table 1.
8. Lines 236-239, very confusing sentence, rewrite with clear explanation. There are several places where this kind of sentence causes confusion.
9. No need to say shown in figure, instead use the figure number directly in parentheses, e.g., (Figure 1, 2, etc.).
10. Line 299, check the temperature written with 0 mM NaCl.
11. So many citations, I think provide important and recent references with precise explanation of result and discussion.

Experimental design

The experimental design is good, but a little confusing because of so many combinations of stress and treatment.

Validity of the findings

Novelty is not addressed well but still, MS has the potential to improve.

Reviewer 2 ·

Basic reporting

This paper does not have any language issues, with standardized writing, appropriate citation of literature, sufficient introduction of research background, strong logical coherence between paragraphs, and the provided charts that fully support the viewpoints and conclusions of the paper. Work is very meaningful.

Experimental design

The experimental design is reasonable, comprehensive, and can effectively support the conclusions and viewpoints of the paper. However, adding a flowchart in the "Materials and Methods" section would be more beneficial for readers to understand.

Validity of the findings

The experimental results are accurate and reliable, and the interpretation of the meaning represented by the results is precise and profound. It is possible to add some recent relevant literature in the discussion section.

Additional comments

Dear authors,
After reviewing the paper entitled “Gamma-aminobutyric acid elicits H2O2 signaling and promotes wheat seed germination under combined salt and high temperature stress (#99354)” by Yu et al., this work reports for the first time a new method that effectively enhances wheat seed germination under high temperature and salt stress using a non protein amino acid, which is of great significance in production and theoretical research. The experimental design for this work is feasible, the data is reliable, and the necessity of the research work is clearly explained in the introduction. The significance and advantages of the experimental results were also discussed in depth. It is indeed a very interesting and meaningful work. However, there are some problems that need to be improved.
1. Abstract. The abstract of the paper needs to be rewritten, paying attention to the structural hierarchy and logical coherence between sentences, and presenting the most important information.
2. Materials and methods. It is recommended to use a flowchart for experimental design, as it will be easier to understand.
3. Discussion. Suggest adding a hypothesis model diagram in the discussion section, which may help readers better understand the significance of the results obtained from this work.
In addition, there are some minor details that need to be corrected, such as the names of genes at lines 408, 411, and 413 that need to be highlighted in italics. Additionally, at the end of the discussion section, a brief description of the shortcomings of this method and future research directions can be provided.

Reviewer 3 ·

Basic reporting

Abstract:The abstract could be more concise. Remove unnecessary details and focus on providing a clear and brief overview of the objectives, methods, results, and conclusions of the study.
GABA, should be written in full-name when mentioned first.
Specify the exact methodology used in the in vitro study (e.g., temperature used, ...) to enhance clarity
Rewrite this sentence, (This study conducted a combined stress experiment of 50 mM NaCl and high temperature (30 °C), using techniques such as high-performance liquid chromatography, the xylenol iron orange method, nitrogen blue tetrazole photochemical reduction, and real-time quantitative PCR to investigate the regulatory effect of GABA on wheat seed germination under salt and high temperature combined stress.).

Based on (200 mM NaCl) or high temperature (30, 35 °C) signiûcantly inhibited wheat seed germination (P < 0.05). Low salt (50 mM NaCl) or an appropriate increase in culture temperature (25 °C) increased the germination rate (GR) of wheat seeds; however, low salt also inhibited GR at high temperature (30 °C).) what is the optimal or high temperature?

Introduction
The introduction should be rewritten include the effect of individual stresses applied, then their combination under field conditions. Then, focus on the physiological changes under combined stress.
The references are too old.
Clearly state the research gap or problem that this study aims to address.
Explicitly state the objectives of the study at the end of the introduction to guide readers on what to expect.
Write on the sensitivity of wheat to stresses applied.

Materials
Provide more details on certain methods.

Results

Present results in a clear and organized manner.
Specify the statistical tests used and include information on significance levels.
Ensure clarity in the presentation of statistical results.
I recommend inserting some photos in your study. It is always better to see the phenotypes.

Discussion

Stay focused on the main findings of the study and avoid introducing too much background information.
Discuss the results in the context of existing literature and explain the mechanisms behind the observed effects.

Conclusion

Briefly summarize the main findings of the study in a clear and concise way.
Highlight the practical implications of the findings, especially for sustainable agriculture.
Language
Some sentences can be shortened or rephrased to be more concise.
Use active voice whenever possible.
Proofread carefully for typos and grammatical errors.
Use consistent formatting throughout the sections.

References

Check the journal outlines of references format.
Double-check all citations to ensure accuracy and consistency. Make sure that references are correctly cited in the text and listed in the bibliography.

Experimental design

Must be improved.
Organize the Materials and Methods section more logically. Group related procedures together and consider using subheadings for different aspects of the methodology.

Validity of the findings

The statistical analysis

In multi-factorial conducted experiments, it is recommended to compare the mean values of data obtained from temperatures, treatments etc.

---

## Round 0.2 · accepted · Accept

I appreciate your constructive attitude toward the suggestions of reviewers and improving your article based on their suggestions. I believe your manuscript is now ready for publication. We look forward to your next article.

·

Basic reporting

I am happy with the current version of MS. The abstract is well written, the methods and results sections are improved. All comments are clearly addressed. Overall, this version is suitable for publication.

Experimental design

Experimental design is well addressed.

Validity of the findings

All the findings are well documented.

Reviewer 2 ·

Basic reporting

The author made excellent revisions to the paper based on the review comments, which greatly improved the language, structure, logic, and best discussion depth of the entire paper. I suggest that the current status of the paper is fully acceptable.

Experimental design

The experimental design is reasonable, especially by adding a flowchart in the revised manuscript to make the experimental design process very clear.

Validity of the findings

The experimental results and conclusions are reliable, and the writing approach for the experimental results is very clear, especially after adding the hypothesis diagram of NADPH oxidase mediated ROS degradation of abscisic acid in the discussion section. Based on the current experimental results, the regulatory mechanism of GABA can be well explained.

Additional comments

Compared to the previous version, this revised version has comprehensively improved the quality and level of the paper, which is very excellent.

Reviewer 3 ·

Basic reporting

The manuscript is highly improved. I recommend acceptance.

Experimental design

It is well presented now.

Validity of the findings

Valid

Additional comments

Nofurther comments